# Factors Associated with Delirium in COVID-19 Patients and Their Outcome: A Single-Center Cohort Study

**DOI:** 10.3390/diagnostics12020544

**Published:** 2022-02-20

**Authors:** Annabella Di Giorgio, Antonio Mirijello, Clara De Gennaro, Andrea Fontana, Paolo Emilio Alboini, Lucia Florio, Vincenzo Inchingolo, Michele Zarrelli, Giuseppe Miscio, Pamela Raggi, Carmen Marciano, Annibale Antonioni, Salvatore De Cosmo, Filippo Aucella, Antonio Greco, Massimo Carella, Massimiliano Copetti, Maurizio A. Leone

**Affiliations:** 1Neurology Unit, Fondazione IRCCS Casa Sollievo della Sofferenza, 71013 San Giovanni Rotondo, Italy; digiorgioannabella@gmail.com (A.D.G.); clara.degennaro@operapadrepio.it (C.D.G.); paoloemilio.alboini84@gmail.com (P.E.A.); luciaflorio81@gmail.com (L.F.); v.inchingolo@gmail.com (V.I.); m.zarrelli@operapadrepio.it (M.Z.); 2Internal Medicine Unit, Fondazione IRCCS Casa Sollievo della Sofferenza, 71013 San Giovanni Rotondo, Italy; a.mirijello@operapadrepio.it (A.M.); s.decosmo@operapadrepio.it (S.D.C.); 3Biostatistics Unit, Fondazione IRCCS Casa Sollievo della Sofferenza, 71013 San Giovanni Rotondo, Italy; a.fontana@operapadrepio.it (A.F.); m.copetti@operapadrepio.it (M.C.); 4Laboratory Medicine Unit, Fondazione IRCCS Casa Sollievo della Sofferenza, 71013 San Giovanni Rotondo, Italy; g.miscio@operapadrepio.it; 5Scientific Research Department, Fondazione IRCCS Casa Sollievo della Sofferenza, 71013 San Giovanni Rotondo, Italy; p.raggi@operapadrepio.it (P.R.); carmen-marciano@libero.it (C.M.); m.carella@operapadrepio.it (M.C.); 6Unit of Clinical Neurology, Department of Neuroscience and Rehabilitation, AOU Sant’Anna, 44124 Ferrara, Italy; annibaleantonioni658@gmail.com; 7Nephrology Unit, Fondazione IRCCS Casa Sollievo della Sofferenza, 71013 San Giovanni Rotondo, Italy; f.aucella@operapadrepio.it; 8Geriatric Unit, Fondazione IRCCS Casa Sollievo della Sofferenza, 71013 San Giovanni Rotondo, Italy; a.greco@operapadrepio.it

**Keywords:** delirium, COVID-19, mortality, C-reactive protein, neutrophils-to lymphocyte ratio

## Abstract

Background: A significant proportion of patients with coronavirus disease 2019 (COVID-19) suffer from delirium during hospitalization. This single-center observational study investigates the occurrence of delirium, the associated risk factors and its impact on in-hospital mortality in an Italian cohort of COVID 19 inpatients. Methods: Data were collected in the COVID units of a general medical hospital in the South of Italy. Socio-demographic, clinical and pharmacological features were collected. Diagnosis of delirium was based on a two-step approach according to 4AT criteria and DSM5 criteria. Outcomes were: dates of hospital discharge, Intensive Care Unit (ICU) admission, or death, whichever came first. Univariable and multivariable proportional hazards Cox regression models were estimated, and risks were reported as hazard ratios (HR) along with their 95% confidence intervals (95% CI). Results: A total of 47/214 patients (22%) were diagnosed with delirium (21 hypoactive, 15 hyperactive, and 11 mixed). In the multivariable model, four independent variables were independently associated with the presence of delirium: dementia, followed by age at admission, C-reactive protein (CRP), and Glasgow Coma Scale. In turn, delirium was the strongest independent predictor of death/admission to ICU (composite outcome), followed by Charlson Index (not including dementia), CRP, and neutrophil-to-lymphocyte ratio. The probability of reaching the composite outcome was higher for patients with the hypoactive subtype than for those with the hyperactive subtype. Conclusions: Delirium was the strongest predictor of poor outcome in COVID-19 patients, especially in the hypoactive subtype. Several clinical features and inflammatory markers were associated with the increased risk of its occurrence. The early recognition of these factors may help clinicians to select patients who would benefit from both non-pharmacological and pharmacological interventions in order to prevent delirium, and in turn, reduce the risk of admission to ICU or death.

## 1. Introduction

Delirium is a neurocognitive disorder characterized by disturbance in attention and awareness that develops over a short period of time (hours, days), fluctuates, and represents a change from the baseline behavioral state as a consequence of an underlying medical condition [1]. Its occurrence is the highest among hospitalized older individuals. Delirium has been associated with high mortality, increased morbidity, functional decline, extended length of hospital stay and increased requirement for institutional care [2]. Features of delirium predicting worse outcomes include older age, frailty, hypoactive subtype, and delirium severity and duration2. Previous studies in patients with coronavirus disease 2019 (COVID-19) demonstrated that 20–30% of patients present with or develop delirium during their hospitalization [3], especially in the elderly [4,5], with rates of 55–70% in cases of severe illness [6]. However, although delirium has emerged as a major complication in the clinical management of COVID-19 patients, there has been a lack of attention paid to its identification, to the ascertainment of its risk factors and to the impact of its outcomes in COVID-19 patients [7]. Reasons for this include the various definitions of delirium, the absence of a systematic assessment for delirium in current COVID-19 management guidelines, the retrospective nature of the studies, the different clinical classification of delirium, the lack of recognition of delirium as an atypical presentation of the disease, and the failure to account for delirium’s impact on mortality [8]. Two were the objectives of the present study: first, to evaluate the occurrence of delirium in a large cohort of adults hospitalized for COVID-19 and its impact on in-hospital mortality; second, to identify risk factors associated with delirium; in this regard, our attention was focused on inflammatory biomarkers. These biomarkers are particularly important, as immunopathology has been suggested as a primary driver of morbidity and mortality with COVID-19 [9].

## 2. Materials and Methods

### 2.1. Design and Sample

This was a single-center observational study conducted at the “IRCCS Casa Sollievo della Sofferenza” hospital, a 900-bed general hospital with a catchment area of about 300,000 inhabitants, in Southern Italy. From 3 March to 31 May 2020, all consecutive patients aged ≥18 years suspected of COVID-19 infection were admitted to our COVID units (internal medicine and geriatric units). All patients had epidemiologic, clinical, laboratory and radiological findings suspected for COVID-19 [10]. Real-time reverse-transcriptase polymerase chain reaction from nasopharyngeal swab was performed in all patients and repeated, in the case of a negative result, as appropriate. The study was performed in accordance with the Declaration of Helsinski of 1964 and all its subsequent amendments and revisions, and it was approved by the local Institutional Ethics Committee (COVID-19-SGR–46/2020). Written informed consent was obtained from all participants or from their next of kin for those incapable of providing the informed consent. Inclusion criteria were: informed consent, age ≥ 18-year-old, and admission to a COVID Unit with suspected or proven SARS-CoV-2 infection. The exclusion criteria were: (1) a previous diagnosis of major psychiatric disorders (schizophrenia spectrum disorders), and (2) the impossibility to evaluate the delirium due to precipitating medical conditions, death, or missing charts.

### 2.2. Patient and Public Involvement

Patients and the public were not involved in any way in the design and conduct of the study, in the choice of outcome measures or in the recruitment.

### 2.3. Data Collection

Data were retrieved from clinical records and entered in an electronic clinical record form. Demographic data (age at admission, gender, education, marital status, employment), type and date of onset of symptoms, date of hospital admission, number and type of comorbidity and chronic medications, baseline clinical data at admission (blood pressure, BMI, Glasgow Coma scale-GCS, symptoms), and smoking status were collected. The Charlson Comorbidity Index [11] was calculated to determine the burden of comorbidity; however, in the time-to-event analyses the dementia score was subtracted from the total score because dementia had been already included as a covariate. ECG (with the measure of QTc), arterial blood gas analysis, chest X-ray, and chest CT were also performed. Laboratory tests were carried out at admission to the emergency department or within 24 h and included: blood cell count, activated partial thromboplastin time, International Normalized Ratio (INR), alkaline phosphatase, total bilirubin, aspartate and ala-nine-aminotransferase, ɣ-glutamyl-transpeptidase, total proteins, creatinine, estimated glomerular filtration rate (eGFR), glycemia, erythrocyte sedimentation rate, triglycerides, cholesterol, lactate-dehydrogenase (LDH), creatine-phosphokinase, troponin, D-dimer, procalcitonin, and C-reactive protein (CRP). The neutrophils-to-lymphocytes ratio (NLR) value was calculated as the absolute periphery neutrophil count divided by the absolute periphery lymphocyte count. Clinical data were collected from the patients and implemented with interviews with relatives or care-givers. The diagnosis of delirium (with the date of onset) involved a two-step process: a brief screening based on the evaluation of nursing and doctors’ daily reports in accordance with the criteria of the 4AT tool (alertness, orientation, attention, and acute change or fluctuating course) [12], followed in the case of positivity by a bedside psychiatric evaluation (A.DG.) based on DSM-5 criteria, in order to confirm or not the diagnosis of delirium. In the case of onset before admission, the date of onset was ascertained through interviews with relatives or caregivers. Once diagnosed, delirium was further clinically classified based upon the predominant motor activity profile into three subtypes: hypoactive, hyperactive, and mixed [13]. The hypoactive and the hyperactive subtypes are characterized by decreased and increased motor activity, respectively, while the mixed subtype by features of both subtypes within short time frames. The following outcomes were retrieved from electronic medical records: dates of hospital discharge, Intensive Care Unit (ICU) admission, or death.

### 2.4. Statistical Analysis

Patients’ baseline demographical and clinical characteristics are reported as median and range or as mean and standard deviation for continuous variables (according to their distribution), and as frequencies and percentages for categorical variables. The normal distribution assumption was tested using Shapiro–Wilk and Kolmogorov–Smirnov tests, as well as by graphical inspection of the Q-Q plot. The baseline variables were compared between patients who developed delirium and those who did not, using the Mann–Whitney test and the chi-square test for crude comparisons, and analysis of covariance (ANCOVA) or binary logistic regression for age- and sex-adjusted comparisons. Time to onset of delirium was calculated from the first COVID-19-related symptom to the onset of delirium, or discharge, whichever came first. Overall survival was defined as the time between admission and death. For subjects who were still alive, survival time was censored at discharge. Survival curves were estimated by using the Kaplan–Meier method and compared by the log-rank test. For time to event endpoints, univariable and multivariable proportional hazards Cox regression models were estimated, and risks were reported as hazard ratios (HR) along with their 95% confidence intervals (95% CI). The proportional hazards assumption was tested using Schoenfeld residuals. A stepwise variables selection was used to build multivariable models. A two-sided *p* value < 0.05 was considered for statistical significance. Univariate Cox regression analyses were adjusted for multiple comparisons using the Bonferroni method. All statistical analyses were performed using the computing environment R (R Development Core Team, version 3.3.2).

## 3. Results

Of the 254 consecutive patients admitted for suspected COVID-19 infection, 22 had a final diagnosis of non-COVID-19 infection and were excluded. Of the remaining 232, there were 169 swab-confirmed diagnoses, 26 antibody-confirmed diagnoses, and 37 patients showed clinical and radiological features of COVID-19 despite negative swabs [10]. For 17 of them the evaluation of delirium was impossible (death before assessment, or missing clinical charts). Thus, we ultimately had 214 patients available for analysis, and 47 (22%) were diagnosed with delirium. In 20 patients, delirium was already present at admission, mostly for 1–2 days, whereas in the remaining 27 patients it developed during hospital stay (1 to 72 days after admission). Twenty-one patients presented the hypoactive type, 15 the hyperactive type, and 11 patients showed the mixed type. Demographics and clinical features at admission are displayed in Table 1. Baseline laboratory findings are shown in Table 2.

In the univariable model, after adjustment for multiple comparisons, the statistically significant predictors of delirium development were: age, GCS, Charlson Index without dementia, dementia, pre-admission use of psychotropic drugs, CRP and NLR (Table 3).

These predictors were then entered in a stepwise multivariable model which retained four independent variables. Presence of dementia was the strongest predictor of delirium, followed by age at admission, CRP, and GCS (Table 4).

Twenty-nine patients with delirium died during hospitalization and 10 were admit- ted to the ICU, compared with 26 and 30 patients without delirium, respectively. The presence of delirium increased the probability of death (Table 5) and the length of hospital stay for patients who survived, whereas the probability of admission to ICU and the length of stay in ICU were not affected (Table 5). To investigate whether delirium was independently associated with a poor outcome, we selected a composite outcome (death/admission to the ICU) and evaluated the probability to attain such an outcome in patients with and without delirium (Figure 1), and the predictive value of clinical and laboratory variables in a univariable model (Table 6). After adjustment for multiple comparisons, age, GCS, Charlson Index without dementia and a number of laboratory variables (hemoglobin, white blood cell count, neutrophils, eGFR, LDH, D-dimer, CRP, NLR) were found to be associated with the composite outcome.

These predictors entered the stepwise multivariable model, which retained delirium as the strongest independent predictor (Table 7) of death/admission to ICU, together with only three other variables: Charlson Index without dementia, CRP and NLR. After dividing delirium by clinical type, the probability of death/admission to ICU was higher for patients with the hypoactive subtype than for those with the hyperactive subtype; patients with the mixed subtype had an intermediate risk (Table 7).

## 4. Discussion

In this observational study on consecutive patients admitted to the emergency department of a general hospital, 22% of 214 patients with COVID-19 infection developed delirium during hospital stay or immediately before admission. Factors independently associated with the risk of delirium included older age, GCS, presence of dementia, and CRP. In turn, delirium was the strongest independent predictor of admission to ICU or intra-hospital death.

*Prevalence:* The prevalence of delirium in COVID-19 patients depends on the study population, diagnostic modality, and setting. It is higher in the ICU setting, ranging from 65 to 80%, than in the general wards [14]. Comparison is made difficult by the variety of definitions of delirium (confusion, altered mental status, acute onset of psychotic symptoms, disorientation, decreased level of consciousness, cognitive dysfunction, and encephalopathy) [15] and of modalities of ascertainment. We defined delirium based on a two-step approach and according to DSM-5, whereas most studies relied on the review of medical records or did not include psychiatric evaluation. We found a prevalence in the range of that observed in COVID-19 patients admitted to general medicine wards (6 to 67%) [14] and similar to the percentage of 23% reported in a meta-analysis of 33 studies on pre-COVID- 19 inpatients [2]. Other series in the same setting reported prevalence ratios of delirium almost three times higher in patients with COVID-19 than in those without [15,16]; however, in these series, the prevalence in non-COVID-19 control patients (range = 5.0–7.7%) seems largely underestimated. Unfortunately, we do not have a comparison series of non-COVID patients, and most other studies are also lacking setting- and age-matched controls.

*Outcomes:* Delirium is independently associated with multiple poor outcomes in non-COVID-19 patients [2], especially in its hypoactive subtype. Delirium was the strongest predictor of a composite outcome (death/admission to the ICU) in our series. In the final multivariable model, adjusting for several other covariates, patients with delirium had an almost fourfold probability of reaching the composite outcome, regardless of other factors. The risk was substantially higher for the hypoactive subtype. Delirium was not included in the majority of studies exploring predictive factors for mortality in COVID-19 patients [17]. A meta-analysis of nine studies [18] showed that the mortality rate of COVID-19 patients with delirium was more than twice the mortality rate of those without delirium. In addition to delirium, only three other factors were identified as predictors of death in COVID-19 patients by our multivariable model: comorbidity measured with the Charlson Index, CRP, and NLR. In COVID-19 patients, comorbidity, age, and male sex were the major predictors of death in two large studies [19,20,21], and three meta-analyses [17,22,23] found CRP and NLR as predictors of severity and intra-hospital death as well.

*Factors associated with delirium:* If delirium has such a strong predictive value for poor outcome in COVID-19 patients admitted to general medicine wards, the search for factors predisposing patients to its occurrence (our second question) is of crucial clinical importance. In this regard, there is only sparse evidence in the literature. In a multicenter study including 817 older patients, significant risk factors included age > 75 years, prior psychoactive medication use, comorbidities, and cognitive impairment or dementia [24]. These predisposing factors for delirium in COVID-19 patients overlap with those observed in the pre-COVID era [2]. We examined several clinical and laboratory features at hospital admission as possibly associated with delirium. After adjustment for multiple comparisons, we found that only age, GCS, dementia, Charlson Index without dementia, pre-admission use of psychotropic drugs, CRP and NLR were associated with delirium in the univariable analysis. In particular, NLR and CRP were associated with an increased risk of delirium per ten units of measurement by 45% and 126%, respectively. In the multivariable analysis, only four variables were retained in the model as independently associated: dementia showed the highest predictive value, followed by age at admission, CRP, and GCS. NLR was excluded because of collinearity with CRP, most likely because both are markers of the same process, systemic inflammation. Additionally, comorbidity was removed from the final model because of collinearity with age and dementia. This result is consistent with previous evidence in the non-COVID-19 literature, which indicates that CRP independently predicts delirium [25,26,27], and with recent findings in COVID-19 patients suggesting an association between CRP levels and delirium occurrence [28]. The possible pathophysiologic mechanisms (direct invasion of the brain, brain hypoxia due to systemic hypoxemia or coagulopathy, cytokine storm and neuroinvasion) [29,30,31] that contribute to the occurrence of delirium among COVID-19 patients, and their mutual interaction, are still unclear. Both CRP and NLR are markers of systemic inflammation, thus supporting the hypothesis that inflammation has a role in the pathophysiology of delirium in COVID-19 patients. Systemic inflammatory mediators cross the blood–brain barrier, activate brain microglia, and initiate neuroinflammation, probably through some blood–brain barrier disruption [29,32]. In this scenario, individuals predisposed to a heightened inflammatory response when exposed to an acute stressor are at increased risk of delirium [26].

*Strengths and Limitations:* Strengths of this study include its external validity because of consecutive patient enrollment and its general hospital setting, a high level of delirium diagnostic certainty because of a two-step evaluation, and the inclusion of different putative risk factors associated with delirium. Three limitations are important to mention: first, some data, including education, living situation and socioeconomic status, were missing from the record. These factors may limit the generalizability of the findings. Second, we analyzed predictive factors for delirium in patients admitted to a general ward, while they may be different in another setting, such as the ICU [6]. Third, our outcome measure combined death and admission to the ICU; more frail patients, such as those with delirium, may be less likely to be considered for admission to the ICU, and this could have underestimated their risk to achieve the outcome.

## 5. Conclusions

In conclusion, we found a few clinical features and inflammatory markers associated with delirium in COVID-19 patients. Although they are far too non-specific to be considered as markers for the occurrence of delirium, they can be easily ascertained at admission and used as a proxy for stratifying the risk of developing delirium in COVID-19 patients. In daily clinical practice, identifying COVID-19 patients at risk of delirium can be very challenging. We believe that the clinical and inflammatory markers identified in this study may help clinicians to select patients who would benefit from both non-pharmacological and pharmacological interventions in order to prevent delirium, and in turn to reduce the risk of admission to ICU or death.

## Figures and Tables

**Figure 1 diagnostics-12-00544-f001:**
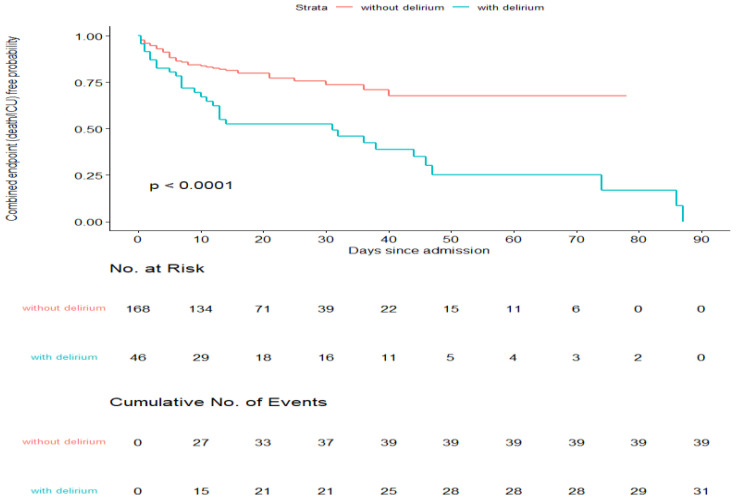
Kaplan–Meier survival curves according to presence/absence of delirium.

**Table 1 diagnostics-12-00544-t001:** Frequency of demographics and clinical characteristics at admission in COVID-19 patients with and without delirium.

	Total (N = 214)	Without Delirium (N = 168)	With Delirium (N = 46)
Age at admission: mean (SD)	67.88 (15.05)	64.32 (14.43)	80.89 (8.89)
Age class (N, %)			
<50	29 (13.6%)	29 (17.3%)	0 (0.0%)
50–59	32 (15.0%)	31 (18.5%)	1 (2.2%)
60–69	44 (20.6%)	40 (23.8%)	4 (8.7%)
70–79	59 (27.6%)	43 (25.6%)	16 (34.8%)
≥80	50 (23.4%)	25 (14.9%)	25 (54.3%)
Male gender (N, %)	118 (55.1%)	97 (57.7%)	21 (45.7%)
Education (years)			
≤5	88 (41.5%)	55 (33.1%)	33 (71.7%)
6–8	55 (25.9%)	48 (28.9%)	7 (15.2%)
>8	69 (32.5%)	63 (38.0%)	6 (13.0%)
Smokers (N, %)	44 (27.2%)	40 (30.5%)	4 (12.9%)
BMI: mean (SD)	26.97 (4.62)	26.98 (4.59)	26.89 (4.84)
Glasgow Coma Scale mean (SD)	14.45 (2.06)	14.71 (1.64)	13.47 (3.00)
Fever at admission (N, %)	154 (72.0%)	128 (76.2%)	26 (56.5%)
Comorbidity at admission (N, %)			
Dementia	33 (15.5%)	9 (5.4%)	24 (52.2%)
Chronic obstructive pulmonary disease	23 (10.7%)	17 (10.1%)	6 (13.0%)
Diabetes	40 (18.7%)	27 (16.1%)	13 (28.3%)
Atrial fibrillation	29 (13.6%)	21 (12.5%)	8 (17.4%)
Arterial hypertension	103 (48.1%)	77 (45.8%)	26 (56.5%)
Hypercholesterolemia	17 (7.9%)	16 (9.5%)	1 (2.2%)
Tumor	40 (18.7%)	33 (19.6%)	7 (15.2%)
Thyreopathy	17 (7.9%)	14 (8.3%)	3 (6.5%)
Hypertriglyceridemia	7 (3.3%)	5 (3.0%)	2 (4.3%)
Others	140 (65.4%)	99 (58.9%)	41 (89.1%)
Charlson Index (without dementia) mean (SD)	3.58 (2.40)	3.15 (2.38)	5.15 (1.71)
Use of psychotropic drugs (N, %)	27 (12.6%)	14 (8.3%)	13 (28.3%)

**Table 2 diagnostics-12-00544-t002:** Baseline † laboratory findings in COVID-19 patients with and without delirium.

	Total (N = 214) ‡	Without Delirium (N = 168) ‡	With Delirium (N = 46) ‡
Red blood cells (RBC) millions/mcl	4.58 (3.97, 5.09)	4.68 (4.13, 5.14)	4.09 (3.70, 4.83)
Hematocrit %	40.25 (35.02, 43.98)	40.80 (36.02, 44.08)	36.25 (32.95, 42.95)
Hemoglobin g/dL	13.30 (11.40, 14.50)	13.40 (11.90, 14.70)	11.80 (10.78, 13.72)
Mean corpuscular volume (MCV) fl	87.80 (84.68, 91.12)	87.30 (83.97, 91.03)	89.20 (86.45, 91.82)
White blood cells (WBC) thousand/mcl	6.39 (4.60, 9.04)	5.92 (4.34, 8.10)	8.29 (5.54, 11.18)
Neutrophils thousand/mcl	4.54 (3.04, 7.68)	4.19 (2.89, 6.39)	6.61 (4.28, 9.27)
Lymphocytes thousand/mcl	1.02 (0.73, 1.45)	1.02 (0.77, 1.46)	1.06 (0.68, 1.42)
Neutrophils-to-lymphocytes ratio (NLR)	4.18 (2.52, 7.41)	3.84 (2.40, 6.52)	6.04 (3.19, 18.89)
Platelets thousand/mcl	212.00 (159.25, 285.50)	205.00 (153.25, 258.00)	273.00 (197.50, 332.25)
International normalized ratio (INR)	1.08 (1.04, 1.15)	1.07 (1.03, 1.13)	1.13 (1.06, 1.19)
Partial thromboplastin time (PTT) sec	24.60 (22.90, 26.90)	24.60 (22.90, 26.90)	24.45 (22.10, 28.00)
Alkaline phosphatase UI/L	74.00 (60.00, 101.50)	71.00 (59.25, 96.75)	85.00 (70.00, 121.00)
Total bilirubin mg/dL	0.60 (0.40, 0.80)	0.60 (0.40, 0.80)	0.60 (0.40, 0.90)
Aspartate amino transferase (AST) UI/L	33.00 (22.00, 48.00)	33.00 (23.00, 48.00)	32.00 (18.50, 47.50)
Alanine amino transferase (ALT) U/L	30.00 (20.00, 48.00)	32.50 (21.00, 51.00)	24.00 (16.00, 38.00)
ɣ-glutamyl-transpeptidase (ɣ-GT) UI/L	46.00 (22.75, 103.00)	51.50 (23.00, 109.50)	35.50 (17.25, 62.75)
Total proteins g/dL	7.00 (6.40, 7.70)	7.15 (6.50, 7.80)	6.70 (6.10, 7.20)
Creatinine mg/dL	0.90 (0.70, 1.10)	0.80 (0.70, 1.10)	1.10 (0.70, 1.60)
Estimated glomerular filtration rate (eGFR), mL/mn	86.00 (58.00, 108.75)	90.00 (66.75, 110.00)	61.50 (33.50, 97.25)
Glycemia mg/dL	106.50 (85.00, 135.00)	104.50 (85.00, 127.75)	120.50 (92.00, 163.00)
Erythrocyte sedimentation rate (ESR) mm	45.00 (30.00, 67.00)	44.00 (29.75, 65.50)	51.00 (42.00, 70.00)
Triglycerides mg/dL	110.00 (88.50, 146.00)	109.00 (84.00, 145.00)	116.00 (106.25, 169.25)
Cholesterol mg/dL	137.00 (111.50, 164.00)	137.00 (112.00, 164.00)	128.50 (105.00, 155.25)
Lactate-dehydrogenase (LDH) U/L	243.00 (194.00, 318.50)	250.50 (196.25, 326.75)	227.00 (180.00, 287.00)
Creatine-phosphokinase CK U/L	83.00 (46.00, 167.00)	80.00 (50.00, 169.00)	88.00 (36.00, 154.00)
Troponin pg/mL	15.10 (8.00, 41.60)	12.60 (7.40, 22.90)	38.85 (16.12, 143.18)
D-dimer mg/mL	764.00 (395.00, 2981.50)	710.00 (393.50, 2136.25)	2345.00 (505.00, 6021.00)
Procalcitonin µg/L	0.14 (0.08, 0.31)	0.12 (0.07, 0.27)	0.23 (0.12, 0.42)
C-reactive protein (CRP) mg/dL	4.93 (1.52, 11.05)	3.59 (1.07, 9.04)	10.70 5.73, 15.75)

† Laboratory tests were performed at the emergency department, or within 24 h after admission; ‡ Values are reported as median (Q1, Q3).

**Table 3 diagnostics-12-00544-t003:** Association between baseline clinical and laboratory variables and the diagnosis of delirium.

			Occurrence of Delirium		
Variable	N †	N Events ‡	Hazard Ratio (95% Confidence Interval)	*p*-Value	*p*-Value Adjusted *
Age ^	214	46	1.11 (1.08–1.15)	1.15 × 10^−10^	5.51 × 10^−9^
Male gender	214	46	0.67 (0.38–1.2)	0.182434625	1
Current/former smokers	162	31	0.42 (0.15–1.21)	0.107783665	1
BMI ^	192	34	1.01 (1–1.02)	0.041076706	1
Glasgow Coma Scale ^	205	43	0.86 (0.79–0.93)	0.00028387	0.013625766
Fever	214	46	0.43 (0.24–0.76)	0.004144349	0.198928741
Chronic obstructive pulmonary disease	214	46	1.21 (0.51–2.87)	6.60 × 10^−1^	1.00
Diabetes	214	46	1.99 (1.04–3.8)	0.036791187	1
Atrial fibrillation	214	46	1.39 (0.65–2.97)	0.402521663	1
Arterial hypertension	214	46	1.48 (0.83–2.66)	0.184784901	1
Hypercholesterolemia	214	46	0.21 (0.03–1.54)	0.125310853	1
Neoplasia	214	46	0.71 (0.32–1.59)	0.401540797	1
Thyreopathy	214	46	0.91 (0.28–2.94)	0.875627181	1
Hypertriglyceridemia	214	46	1.32 (0.32–5.47)	0.705744187	1
Other comorbidities	214	46	5.32 (2.09–13.51)	0.000439886	0.02111453
Charlson Index (without dementia) ^	214	46	1.35 (1.2–1.53)	6.39966 × 10^−7^	3.07184 × 10^−5^
Use of psychotropic drugs	214	46	3.58 (1.87–6.83)	0.000112691	0.005409186
Dementia	213	46	11.21 (6.13–20.5)	4.17 × 10^−15^	2.00 × 10^−13^
Vit. D therapy	214	46	4.28 (1.68–10.91)	2.34 × 10^−3^	0.11250987
Neutrophil/lymphocyte ratio ^§^	210	44	1.45 (1.21–1.73)	4.87904 × 10^−5^	0.002341939
Hematocrit ^§^	210	44	0.67 (0.43–1.03)	0.067830978	1
Hemoglobin ^§^	210	44	0.18 (0.05–0.69)	0.012064584	0.579100049
MCV ^§^	204	40	1.36 (0.88–2.1)	0.160177209	1
RBC ^§^	210	44	0.28 (0.01–9.86)	0.480843162	1
WBC ^§^	210	44	1.35 (1.02–1.79)	0.03343282	1
Neutrophils ^§^	210	44	1.11 (0.93–1.31)	0.243821628	1
Lymphocytes ^§^	210	44	0.22 (0.01–4.87)	0.337755457	1
Platelets ^§^	210	44	1.04 (1.01–1.07)	0.002220278	0.106573321
INR ^§^	194	38	6.74 (0.15–310.51)	0.32861737	1
PTT ^§^	194	38	1.13 (0.61–2.1)	0.703355655	1
ALT ^§^	205	43	0.95 (0.87–1.04)	0.244626107	1
Total bilirubin ^§^	202	41	2.81 (0.07–110.22)	0.581749862	1
Alkaline phosphatase ^§^	163	37	1.03 (0.98–1.08)	0.23127989	1
ɣ-GT ^§^	164	36	1 (0.97–1.03)	0.820668706	1
AST ^§^	205	43	0.97 (0.9–1.05)	0.44704993	1
Total proteins ^§^	203	41	0.03 (0–1.05)	0.053086568	1
Creatinine ^§^	206	43	1.05 (0.4–2.78)	0.923433337	1
e-GFR ^§^	202	42	0.9 (0.83–0.97)	0.004767905	0.228859456
Glycemia ^§^	204	42	1.06 (1.03–1.11)	0.000990944	0.047565326
ESR ^§^	149	25	1.1 (0.95–1.27)	0.184748898	1
Triglycerides ^§^	163	30	1.05 (0.99–1.11)	0.11199928	1
Cholesterol ^§^	163	30	0.97 (0.89–1.07)	0.59449302	1
LDH §	155	31	0.99 (0.96–1.02)	0.623247956	1
CK §	162	37	1 (0.99–1.02)	0.856646053	1
Troponin §	101	24	1 (0.99–1.01)	0.750801089	1
D-dimer §	119	25	1 (1–1)	0.029007362	1
Procalcitonin §	118	24	1.75 (0.89–3.47)	1.06 × 10^−1^	1
CRP §	207	43	2.26 (1.6–3.2)	4.28496 × 10^−6^	0.000205678

† Number of subjects with information available; ‡ number of events (persons with delirium); * Bonferroni adjustment; ^ HR per 1 unit increment; § HR per 10 units increment (units for each variable are listed in Table 2).

**Table 4 diagnostics-12-00544-t004:** Variables associated with delirium occurrence in the multivariable model.

	Occurrence of Delirium	
Variable	Hazard Ratio (95% Confidence Interval)	*p*-Value
Age at admission ^	1.07 (1.03–1.11)	0.0007
Glasgow coma scale	0.88 (0.8–0.98)	0.0166
Baseline CRP §	1.06 (1.02–1.1)	0.0015
Dementia	3.2 (1.38–7.38)	0.0065

^ HR per 1 unit increment; § HR per 10 units increment.

**Table 5 diagnostics-12-00544-t005:** Association between delirium and outcomes.

	Patients with Delirium (N = 46)	Patients without Delirium (N = 168)	*p*-Value
Length of hospitalization (only survived patients days) (mean, SD)	33.44 (12.63)	24.64 (13.28)	Poisson model, *p* < 0.0001
Admission to ICU (N, %)	10/46 21.7%	30/168 17.9%	Cox model Hazard Ratio = 1.26, *p* = 0.53
Time to ICU admission (only patients admitted in ICU) days (mean, SD)	10.60 (15.04)	6.90 (7.40)	Poisson model, *p* < 0.0005
Length of stay in ICU (only patients admitted in ICU) days (mean, SD)	24.00 (4.24)	24.00 (8.73)	Poisson model, *p* = 0.893
In-hospital death (N, %)	29/46 63.0%	26/168 15.5%	Cox model Hazard Ratio = 8.27, *p* < 0.0001

**Table 6 diagnostics-12-00544-t006:** Association between baseline clinical and laboratory variables and the combined endpoint (death or admission to ICU).

Combined Endpoint (Death/Admission to ICU)					
Variable	N †	N Events ‡	Hazard Ratio (95% Confidence Interval)	*p*-Value	*p*-Value Adjusted *
Delirium (time dependent)	214	70	5.12 (2.99–8.78)	0.000000003	0.0000001
Age at admission ^	214	70	1.05 (1.03–1.08)	0.000000237	0.0000116
Male gender	214	70	1.05 (0.65–1.69)	0.837643647	1.0000000
Current/former smokers	162	50	1.12 (0.59–2.13)	0.722477102	1.0000000
BMI ^	192	58	1.01 (1–1.01)	0.121131099	1.0000000
Glasgow Coma Scale ^	205	65	0.86 (0.8–0.93)	0.000145628	0.0071358
Fever	214	70	0.69 (0.42–1.13)	0.142083876	1.0000000
Chronic Pulmonary Obstructive Disease	214	70	2.15 (1.19–3.88)	0.011525366	0.5647430
Diabetes	214	70	1.39 (0.8–2.41)	0.246348210	1.0000000
Atrial fibrillation	214	70	1.65 (0.91–2.97)	0.096134852	1.0000000
Arterial hypertension	214	70	1.7 (1.05–2.76)	0.030715219	1.0000000
Hypercholesterolemia	214	70	0.47 (0.15–1.5)	0.200883917	1.0000000
Tumor	214	70	1.74 (1.02–2.95)	0.041879511	1.0000000
Thyreopathy	214	70	1.19 (0.51–2.76)	0.684096031	1.0000000
Hypertriglyceridemia	214	70	0.79 (0.19–3.21)	0.738138639	1.0000000
Other comorbidities	214	70	2.65 (1.45–4.84)	0.001604357	0.0786135
Charlson Index (without dementia) ^	214	70	1.28 (1.16–1.41)	0.000000394	0.0000193
Use of psychotropic drugs	214	70	1.78 (0.97–3.25)	0.062649518	1.0000000
Dementia	213	69	2.36 (1.39–4.03)	0.001521415	0.0745493
Vit. D therapy	214	70	1.21 (0.38–3.85)	0.750983797	1.0000000
Neutrophil/lymphocyte ratio §	210	67	1.57 (1.37–1.81)	0.000000000	0.0000000
Hematocrit §	210	67	0.62 (0.44–0.89)	0.009681733	0.4744049
Hemoglobin §	210	67	0.15 (0.05–0.45)	0.000721321	0.0353447
MCV §	204	62	1.24 (0.89–1.73)	0.202686714	1.0000000
RBC §	210	67	0.07 (0–1.61)	0.096831372	1.0000000
WBC §	210	67	1.55 (1.24–1.94)	0.000139079	0.0068149
Neutrophils §	210	67	1.22 (1.09–1.37)	0.000454567	0.0222738
Lymphocytes §	210	67	0.99 (0.54–1.8)	0.972133027	1.0000000
Platelets §	210	67	1 (0.98–1.03)	0.880538484	1.0000000
INR §	194	60	10.95 (0.63–189.64)	0.099877021	1.0000000
PTT §	194	60	1.52 (0.96–2.41)	0.075694508	1.0000000
ALT §	205	65	1.01 (0.97–1.04)	0.702865821	1.0000000
Total bilirubin §	202	62	3.29 (0.19–55.53)	0.408924418	1.0000000
Alkaline phosphatase §	163	52	1.05 (1.01–1.09)	0.014802566	0.7253258
ɣ-GT §	164	53	1.01 (0.99–1.03)	0.329223790	1.0000000
AST §	205	65	1.02 (1–1.05)	0.071553567	1.0000000
Total proteins §	203	63	0.02 (0–0.3)	0.005466919	0.2678790
Creatinine §	206	65	1.36 (0.81–2.3)	0.250416795	1.0000000
e-GFR §	202	65	0.89 (0.83–0.95)	0.000202712	0.0099329
Glycemia §	204	64	1.05 (1.02–1.08)	0.002704169	0.1325043
ESR §	149	42	1.08 (0.96–1.2)	0.184670429	1.0000000
Triglycerides §	163	49	1 (0.95–1.05)	0.994884031	1.0000000
Cholesterol §	163	49	0.93 (0.86–1.01)	0.087372174	1.0000000
LDH §	155	45	1.03 (1.02–1.05)	0.000171386	0.0083979
CK §	162	55	1 (0.99–1.02)	0.578550518	1.0000000
Troponin §	101	29	1 (1–1)	0.033795411	1.0000000
D-dimer §	119	35	1 (1–1)	0.000014371	0.0007042
Procalcitonin §	118	38	1.02 (0.49–2.16)	0.949739846	1.0000000
CRP §	207	65	2.37 (1.76–3.19)	0.000000013	0.0000006

† Number of subjects with information available; ‡ number of events (persons with death/admission to ICU); ***** Bonferroni adjustment; ^ HR per 1 unit increment; § HR per 10 units increment (units for each variable are listed in Table 2).

**Table 7 diagnostics-12-00544-t007:** Predictors of poor outcome (death or admission to ICU) in the multivariable model.

	Model A		Model B	
Variable	Hazard Ratio (95% Confidence Interval)	*p*-Value	Hazard Ratio (95% Confidence Interval)	*p*-Value
Neutrophils-to-lymphocytes ratio §	1.03 (1.01–1.05)	0.001443974	1.03 (1.01–1.05)	0.001066611
Baseline CRP §	1.07 (1.03–1.1)	0.000282734	1.06 (1.02–1.1)	0.001314788
Charlson Index (without dementia)	1.21 (1.07–1.36)	0.001655282	1.22 (1.08–1.38)	0.000977608
Delirium	3.81 (2.15–6.73)	4.25597 × 10^−6^		
Hyperactive delirium (=15)			2.09 (0.64–6.83)	0.2199668
Hypoactive delirium (N = 21)			5.95 (2.85–12.42)	2.05376 × 10^−6^
Mixed delirium (N = 11)			3.02 (1.08–8.46)	0.03588895

Reference category is no delirium. Model A: any type of delirium; Model B: separately evaluates hypo-, hyperactive or mixed; § HR for 10 units increment (units for each variable are listed in Table 2).

## Data Availability

The data presented in this study are available on request from the corresponding author. The data are not publicly available due to ethical and privacy restrictions.

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
