# Peer review of "Factors Associated with Delirium in COVID-19 Patients and Their Outcome: A Single-Center Cohort Study"

_diagnostics, 2022, doi:10.3390/diagnostics12020544_

Round 1

Reviewer 1 Report

I found this research article well designed, well-executed and well written. Topic selected for this study "Factors associated with delirium in COVID-19 patients and their outcome: A single-center cohort study" is also interesting and contemporary.

Within the result section I found some data presentation problem therefore first paragraph of page 4 must be reviewed again [Of the 254 consecutive patients admitted for suspected COVID-19 infection, swab confirmed diagnoses were 169, antibody-confirmed diagnoses were 26, and 37 patients showed clinical and radiological features of COVID-19 despite negative swabs. Forty patients were excluded due to other diagnoses (22) or impossible evaluation of delirium (17). Thus, 214 patients were considered for the analysis, and 47 (22%) were diagnosed with delirium].

Kindly correct  typo error within Table 2 at page 5 [erythrocyte sedimentation rate].

Overall I found this article interesting and I recommend it for publication.

Author Response

We wish to thank the reviewer for his/her appreciation of our work.

  • Within the result section I found some data presentation problem therefore first paragraph of page 4 must be reviewed again [Of the 254 consecutive patients admitted for suspected COVID-19 infection, swab confirmed diagnoses were 169, antibody-confirmed diagnoses were 26, and 37 patients showed clinical and radiological features of COVID-19 despite negative swabs. Forty patients were excluded due to other diagnoses (22) or impossible evaluation of delirium (17). Thus, 214 patients were considered for the analysis, and 47 (22%) were diagnosed with delirium].

Response: For the sake of clarity, we have modified the sentence as follows: “Of the 254 consecutive patients admitted for suspected COVID-19 infection, 22 had a final diagnosis of non-COVID-19 infection and were excluded. Of the remaining 232, swab-confirmed diagnoses were 169, antibody-confirmed diagnoses were 26, and 37 patients showed clinical and radiological features of COVID-19 despite negative swabs10.  For 17 of them evaluation of delirium was impossible (death before assessment, or missing clinical charts). Thus, we ended with 214 patients available for the analysis, and 47 (22%) were diagnosed with delirium”. 

  • Kindly correct typo error within Table 2 at page 5 [erythrocyte sedimentation rate].

Response: Thanks, done.

Reviewer 2 Report

Dear Authors

This cohort study examined the occurrence of delirium, the associated risk factors, and the impact on the prognosis of patients with COVID-19. It is a timely theme and an interesting research paper. The particular focus of this paper on inflammatory biomarkers is valuable, considering its relevance to COVID-19. The topic aligns well with the theme of this journal (special issue) and it could be an interesting paper for readers of this journal. There seems to be no problem with the sample or the study method. However, I would like to suggest some minor corrections, which I have outlined below. 

In the “Abstract” section, the authors described “The probability to reach the outcome was higher …”

Does this “outcome” refer to the “worst outcome” or “poor outcome” found elsewhere? If so, please add modifiers as appropriate.

In the “Abstract” section, the authors described “The early recognition of these factors may enable …”

Please explain how the results of this research can be effectively applied to medical institutions and clinical practice. For example, please share your thoughts regarding the effective utilization of the results.

In the “Keywords” section, “PCR” is listed as one of the keywords. However, this abbreviation does not appear in the text. Please ensure that the abbreviation “PCR” appears in the text as well.

In the “2.3. Data collection” section, the authors described “a bedside psychiatric evaluation (A.D.G.) based on the DSM-5 criteria …” Please clarify the meaning of the abbreviation “A.D.G.”.

In the “2.3. Data collection” section, the authors described “The hypoactive and the hyperactive subtypes characterized by increased and decreased motor activity respectively …”

The order seems to be reversed (hypoactive/hyperactive, increased/decreased). Please verify this and amend as necessary.

In the “Results” section, the authors described “Of the 254 consecutive patients admitted for suspected COVID-19 infection, swab-confirmed diagnoses were 169, antibody-confirmed diagnoses were 26, and 37 patients showed …”

The breakdown of 254 patients is unclear. Please specify what 169, 26, 37 are referring to. Also, should the sum of these numbers add up to 254 in total? Please make the appropriate corrections.

In the “Results” section, the authors described “Forty patients were excluded due to other diagnoses (22) or impossible evaluation of delirium (17) …”

I do not understand the inclusion of (22) and (17). Please clarify.

In “Table 3”, the symbol (§) was used but not explained. Please add an explanation in the footnote of the table.

In “Table 2” and the “Discussion” sections, the authors used the abbreviation “ED”.

Please clarify the meaning of the abbreviation “ED”.

In the “Discussion” section, the authors described “patients (5.0-7,7%) seems…” Please make appropriate corrections.

In the “Discussion” section, the authors described “… found CRP and NLS as predictors …”

Please clarify the meaning of the abbreviation “NLS”.

Please add a line of space between the “Conclusions” section and the “Author Contributions” section.

Author Response

We wish to thank the reviewer for his/her appreciation of our work

1) In the “Abstract” section, the authors described “The probability to reach the outcome was higher .. Does this “outcome” refer to the “worst outcome” or “poor outcome” found elsewhere? If so, please add modifiers as appropriate.

Response: Thanks for raising this issue. We have now better explained the meaning.

2) In the “Abstract” section, the authors described “The early recognition of these factors may enable …” Please explain how the results of this research can be effectively applied to medical institutions and clinical practice. For example, please share your thoughts regarding the effective utilization of the results.

Response: We thank the reviewer for the suggestion to clarify the clinical utility of our results.  We have modified the relative sentences in the discussion and abstract.

3) In the “Keywords” section, “PCR” is listed as one of the keywords. However, this abbreviation does not appear in the text. Please ensure that the abbreviation “PCR” appears in the text as well.

Response: Sorry for the uncorrected abbreviation. It has been changed throughout the text with C-reactive protein.

 4) In the “2.3. Data collection” section, the authors described “a bedside psychiatric evaluation (A.D.G.) based on the DSM-5 criteria …” Please clarify the meaning of the abbreviation “A.D.G.”.

Response: A.D.G stands for Annabella Di Giorgio, the first author who did the psychiatric assessment.

 5) In the “2.3. Data collection” section, the authors described “The hypoactive and the hyperactive subtypes characterized by increased and decreased motor activity respectively …” The order seems to be reversed (hypoactive/hyperactive, increased/decreased). Please verify this and amend as necessary.

Response: Thanks, you are right.  We have now amended it.

 6) In the “Results” section, the authors described “Of the 254 consecutive patients admitted for suspected COVID-19 infection, swab-confirmed diagnoses were 169, antibody-confirmed diagnoses were 26, and 37 patients showed …” The breakdown of 254 patients is unclear. Please specify what 169, 26, 37 are referring to. Also, should the sum of these numbers add up to 254 in total? Please make the appropriate corrections.

Response: For the sake of clarity, we have modified the sentence as follows: “Of the 254 consecutive patients admitted for suspected COVID-19 infection, 22 had a final diagnosis of non-COVID-19 infection and were excluded. Of the remaining 232, swab-confirmed diagnoses were 169, antibody-confirmed diagnoses were 26, and 37 patients showed clinical and radiological features of COVID-19 despite negative swabs.  For 17 of them evaluation of delirium was impossible (death before assessment, or missing clinical charts). Thus, we ended with 214 patients available for the analysis, and 47 (22%) were diagnosed with delirium”.  

 7) In the “Results” section, the authors described “Forty patients were excluded due to other diagnoses (22) or impossible evaluation of delirium (17) …” I do not understand the inclusion of (22) and (17). Please clarify.

Response: We have now better explained this issue. Please, see point 6.

 8) In “Table 3”, the symbol (§) was used but not explained. Please add an explanation in the footnote of the table.

Response: The symbol (§) was already explained. Please, check in the footnote § for 10 units increment (units for each variable are listed in Table 2).

 9) In “Table 2” and the “Discussion” sections, the authors used the abbreviation “ED”. Please clarify the meaning of the abbreviation “ED”.

Response: Thanks for this advice. The abbreviation ED was used as acronym for Emergency Department. We have now changed it with the full-length words throughout the text and in Table 2.

 10) In the “Discussion” section, the authors described “patients (5.0-7,7%) seems…” Please make  appropriate corrections.

        Response: Sorry for the typo. We have corrected it.

 11) In the “Discussion” section, the authors described “… found CRP and NLS as predictors …”  Please clarify the meaning of the abbreviation “NLS”.

        Response: Sorry for the wrong acronym. We have changed NLS with NLR (neutrophil-to-lymphocyte ratio).

12) Please add a line of space between the “Conclusions” section and the “Author Contributions”   section.

       Response: Thanks, done.